# The impact of ultrasound-based antenatal screening strategies to detect vasa praevia in the United Kingdom: An exploratory study using decision analytic modelling methods

Benjamin Ruban-Fell[1], George Attilakos[2], Tao Haskins-Coulter[1], Christopher Hyde[3], Jeanette Kusel[1], Anne Mackie[4], Oliver Rivero-Arias[5], Basky Thilaganathan[6], Nigel Thomson[7], Cristina Visintin[8], John Marshall[8]*

1 Costello Medical, London, United Kingdom, 2 Fetal Medicine Unit, University College London Hospital, London, United Kingdom, 3 Exeter Test Group, Institute of Health Research, College of Medicine and Health, University of Exeter, St. Luke's Campus, Exeter, United Kingdom, 4 National Screening Committee, Public Health England, London, United Kingdom, 5 National Perinatal Epidemiology Unit, Nuffield Department of Population Health, University of Oxford, Oxford, United Kingdom, 6 Fetal Medicine Unit, St George's University Hospital NHS Foundation Trust and Molecular & Clinical Sciences Research Institute, St George's University of London, London, United Kingdom, 7 The Society and College of Radiographers, London, United Kingdom, 8 UK National Screening Committee, London, United Kingdom

* john.marshall@dhsc.gov.uk

## Abstract

The objective of this exploratory modelling study was to estimate the effects of second-trimester, ultrasound-based antenatal detection strategies for vasa praevia (VP) in a hypothetical cohort of pregnant women. For this, a decision-analytic tree model was developed covering four discrete detection pathways/strategies: no screening; screening targeted at women undergoing in-vitro fertilisation (IVF); screening targeted at women with low-lying placentas (LLP); screening targeted at women with velamentous cord insertion (VCI) or a bilobed or succenturiate (BL/S) placenta. Main outcome measures were the number of referrals to transvaginal sonography (TVS), diagnosed and undiagnosed cases of VP, over-detected cases of VCI, and VP-associated perinatal mortality. The greatest number of referrals to TVS occurred in the LLP-based (2,083) and VCI-based screening (1,319) pathways. These two pathways also led to the highest proportions of pregnancies diagnosed with VP (VCI-based screening: 552 [78.9% of all pregnancies]; LLP-based: 371 [53.5%]) and the lowest proportions of VP leading to perinatal death (VCI-based screening: 100 [14.2%]; LLP-based: 196 [28.0%]). In contrast, the IVF-based pathway resulted in 66 TVS referrals, 50 VP diagnoses (7.1% of all VP pregnancies), and 368 (52.6%) VP-associated perinatal deaths which was comparable to the no screening pathway (380 [54.3%]). The VCI-based pathway resulted in the greatest detection of VCI (14,238 [99.1%]), followed by the IVF-based pathway (443 [3.1%]); no VCI detection occurred in the LLP-based or no screening pathways. In conclusion, the model results suggest that a targeted LLP-based approach could detect a substantial proportion of VP cases, while avoiding VCI overdetection and requiring minimal changes to current clinical practice. High-quality data is required to

**Data Availability Statement:** All relevant data are included within the paper and supplementary information.

**Funding:** This study was funded by the UK NSC. The views and opinions expressed by the authors in this publication are not necessarily those of the UK NSC.

**Competing interests:** All authors have completed the ICJME uniform disclosure form at www.icmje. org/coi_disclosure.pdf (available on request from the corresponding author) and declare: no support from any organisation for the submitted work other than that described above; JM, CV, and AM are employees of the UK NSC secretariat which funded the submitted work; CH is a member of the UK NSC; BT and ORA are members of the Fetal, Maternal and Child Health Group (FMCH) of the UK NSC; GA is a Council member of the Royal College of Obstetricians and Gynaecologists and a Steering Committee member of the UK Obstetric Surveillance System; BRF, JK and THC are, or were formerly, employed by Costello Medical which was commissioned for the model development and supportive work by the UK NSC; no other relationships or activities that could appear to have influenced the submitted work. The above statement does not alter our adherence to PLOS ONE policies on sharing data and materials.

explore the clinical and cost-effectiveness of this and other detection strategies further. This is necessary to provide a robust basis for future discussion about routine screening for VP.

## Introduction

Vasa praevia (VP) is a rare condition whereby fetal blood vessels run across or close to the cervical opening during labour [1]. Without antenatal detection and intervention through planned Caesarean section, fatal exsanguination of the fetus may occur [2]. Type I VP arises from velamentous cord insertion (VCI) and Type II arises as a consequence of a bilobed or succenturiate (BL/S) placenta [3–5]. A third type has recently been described in cases with abnormal placental location [6].

The development of guidelines for antenatal detection of VP is dependent on a limited evidence base and diagnostic criteria continue to evolve [5, 7]. Strategies vary, but most rely on detecting predisposing risk factors such as low-lying placenta (LLP), BL/S placenta, VCI or, less frequently, marginal cord insertion (MCI) via transabdominal sonography (TAS) in the second trimester, with the presence of VP (and the need for a Caesarean section) confirmed with further TAS and/or transvaginal sonography (TVS) [7–11].

In the UK there is no nationally recommended strategy for antenatal detection and management of VP from clinical guideline bodies [5, 12]. The UK National Screening Committee (UK NSC) does not recommend universal screening for VP. This is based on a review which identified a weak evidence base relating to screening for VP and concerns regarding unnecessary Caesarean sections and VCI overdetection [1].

Systematic detection of VCI, for example as part of a screening strategy for VP, would represent a departure from UK clinical practice as this and other cord anomalies are not included in the panel of mid trimester screening targets [13]. Though VCI has a demonstrated high prevalence in cases with VP [14], and many women with VP will therefore have VCI, only around 2% of women with VCI will also have VP. This suggests a possibly high rate of VCI overdetection if this marker is used to identify a group of women who would be offered further testing for VP. At the same time VCI itself is reported to have an association with a number of adverse perinatal outcomes, albeit weak-to-moderate [15]. Management pathways for VCI based on enhanced monitoring are beginning to be described in guidelines outside the UK [10, 16], which may indicate a growing awareness of this association. However, test accuracy studies are limited and there is an absence of evidence-based interventions for VCI and related anomalies such as MCI [15].

Antenatal VP detection practice and awareness of risk factors have been reported to vary across UK maternity units [17]. While interest in this area is increasing in the UK, a very limited body of UK-based research is available to inform discussion or quantify outcomes from screening strategies [18–20].

The concept of 'screening' is centrally concerned with the early detection of a disease, or risk of disease, in whole populations in which the prevalence of the condition in question is low. The aim of this strategy is to improve outcomes while minimising any screening-related harms from, for example, false positive results, findings of uncertain clinical significance, overdiagnosis or unnecessary interventions. Guidance on VP by the Royal Australian and New Zealand College of Obstetricians and Gynaecologists (RANZCOG) characterises TAS-based screening for VCI as universal, or population, screening strategy [9]. In this screening approach, all pregnant women would be offered TAS for VCI in order to establish the risk of VP. Where the presence of VP is confirmed by TVS diagnosis, women could be offered Caesarean section to prevent the adverse consequences with this condition.

However, discussion of the concept of screening has also identified alternative approaches such as 'targeted screening' [21, 22]. This might be described as a testing intervention which is proactively offered to a group of people identified as being at elevated risk of a condition compared to the general population; an important consequence of this approach is a lower number needing to be screened to detect a case of disease compared to universal (population) screening [23, 24]. LLP stands out as a candidate for such an approach, where placental localisation at mid-term to establish risk of placenta praevia has been embedded in antenatal care for many years and LLP is detected in approximately 10% of pregnancies [13]. Detection and management are well served by guidelines from national bodies, recommending that women with LLP at mid-term should be recalled for further scanning in the third trimester and offered caesarean section where indicated [5, 12].

Given the limited availability of UK evidence on VP, its detection and management, a screening impact model was developed within an expert group. Rather than providing a definitive analysis of all possible pathways based on all possible combinations of risk factors, the aim of this exploratory study was to use decision analytic modelling techniques to develop a series of screening pathways based on discrete risk factors relevant to the UK setting in order to explore the evidence base and to compare the potential impact of each pathway on key outcomes relating to VP. The overall purpose of this work was to make a practical contribution to the evolving discussion about the antenatal detection of VP in the UK; this was achieved by presenting here an analysis of four possible detection pathways for VP which increase in scale and by highlighting the need for high-quality data in order to fully explore the clinical and cost-effectiveness of potential detection strategies in the UK setting.

## Methods

### Model structure

The VP screening model was programmed in Microsoft Excel and used a decision-analytic tree structure to explore the effects of four potential detection pathways in a hypothetical one-year UK pregnancy cohort. Decision trees are appropriate for modelling the short-term outcomes of antenatal screening programmes when these outcomes are based on well-defined processes, such as those assessed in this study [25, 26]. Decision tree structures have been used in previously published VP screening models and in other models of antenatal screening scenarios in a UK population [27–29]. The structure of the modelled decision tree is outlined in Table 1 and S1 Fig.

As part of this exploratory model, each detection pathway was assessed as a discrete decision alternative. During the first stage of the decision tree, the hypothetical pregnancy cohort entered one of four alternative detection pathways; these were designed through expert discussion during two independent workshops and consultation of existing guidelines for VP detection in the US, Australia, New Zealand and Canada [8–10, 30]. The pathways, considered to be of most interest in this exploratory analysis, were: no screening, in-vitro fertilisation (IVF)-based screening, LLP-based screening or VCI-based screening. An overview diagram comparing the different pathways (and their hypothetical integration into clinical practice) is provided in Fig 1, with all four pathways including the recommended $18^{+0}$ to $20^{+6}$ week fetal anomaly scan as a first step. The no screening pathway was designed to provide an approximation of VP detection in current routine clinical practice (in the absence of a nationally recommended VP detection strategy). In this pathway, it was assumed that only pregnancies in which VP was incidentally detected during the $18^{+0}$ to $20^{+6}$ week fetal anomaly scan, as the main component of this pathway, were referred to TVS for VP for confirmation. In all four pathways it was assumed that all pregnancies were examined for LLP during the $18^{+0}$ to $20^{+6}$ week scan, and

**Table 1. Branches of the VP screening model.**

| Section | Description |
|---------|-------------|
| Detection pathways (as the decision alternatives) | One of the four detection pathways is selected at the initial stage |
| Test eligible groups (as per detection pathway) | The group of pregnancies eligible for testing for VP and therefore entering the respective screening pathway are identified at this stage (e.g. the number/ proportion of pregnancies with LLP, IVF or the whole cohort) |
| True (biological) health state | This represents the underlying biological health state of each pregnancy, irrespective of the eventual diagnosis (VCI, VP or uncomplicated pregnancy) |
| Screening result | This segment determines whether VP is diagnosed by TVS or not (and whether a woman is referred to TVS, not referred to TVS or opts out of testing completely). Women may also be re-scanned where TVS is indeterminate for VP diagnosis. Women in the not screened arm may also be diagnosed with VP via TVS, accounting for any incidental diagnoses |
| Birth method | This stage determines whether the birth is planned vaginal or via planned Caesarean section, followed by whether the birth happened as planned or if an emergency Caesarean section was required |
| Survival of the baby | This considers the risk of death at any point in the perinatal period |

**Abbreviations:** IVF, in vitro fertilisation; LLP, low-lying placenta; TVS, transvaginal sonography; VCI, velamentous cord insertion; VP, vasa praevia.

women with LLP were offered a follow-up examination at 32 weeks, in keeping with currently recommended good clinical practice [13]. However, only in the LLP-based pathway was detection of LLP at $18^{+0}$ to $20^{+6}$ weeks followed up with an additional TAS specifically for VP at 32 weeks (representing a targeted screening strategy in which VP is actively sought only in women who have a risk factor routinely detected in current practice to prevent adverse outcomes from placenta praevia). In the VCI-based pathway, additional testing for VCI and BL/S placenta specifically aimed at establishing the risk of VP would be performed during the $18^{+0}$ to $20^{+6}$ week scan, with positive detection prompting a recall at 32 weeks to perform further TAS to confirm the presence of VP (representing a population screening strategy based on a risk factor which is sought for the sole purpose of identifying and preventing adverse outcomes from VP, and which is not currently reported in UK practice). The same strategy was followed for the IVF-based pathway, except that this was applied only to women with pregnancies resulting from IVF; as this is a predisposing factor associated with VP in a very small population subgroup, this was used to represent risk assessment in routine clinical care [4]. All detection pathways focused only on singleton pregnancies, a simplifying assumption based on findings indicating that there is no independently significant association between multiple pregnancies and VP incidence [1, 4]. In all pathways, pregnancies that underwent TAS at 32 weeks were also followed-up by TVS for VP, if VP was suspected. Incidental detection of VP across all pregnancies was also accounted for in all four pathways.

## Data sources

The majority of model inputs were derived from published sources identified through the previously conducted UK NSC review of VP screening [1], targeted literature searches, a systematic literature review (SLR) and meta-analysis (MA) of adverse outcomes associated with (un-) diagnosed VP, as well as normal pregnancies and pregnancies with VCI (S1 File). Quality assessments were conducted for all published sources included in the base case model using the Joanna Briggs Institute Critical Appraisal Checklist for Studies Reporting Prevalence Data for epidemiological studies [31], Centre for Evidence Based Medicine Prognostic Studies

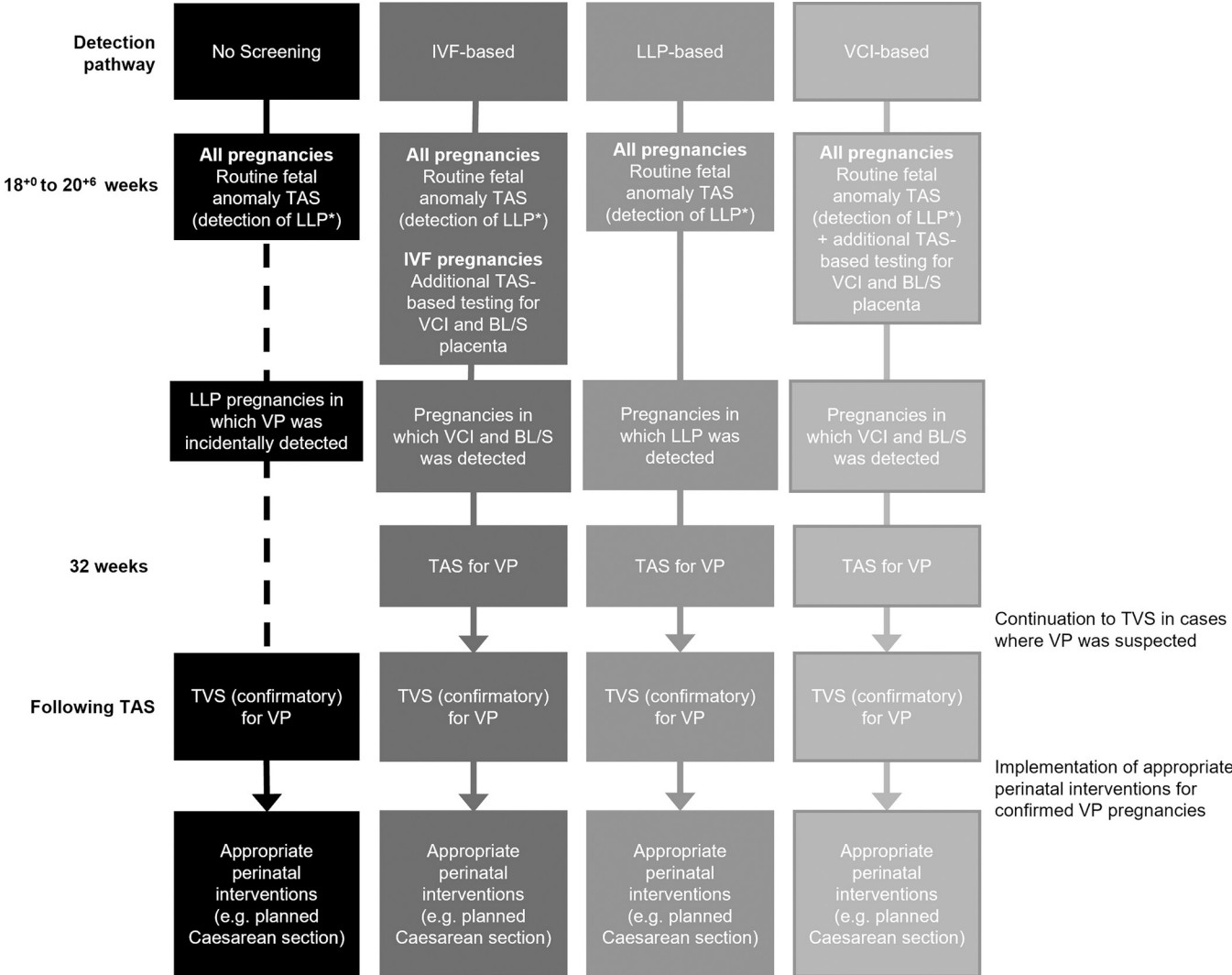

**Fig 1. Modelled detection pathways.** * Women with LLP defined by a placental edge that is 2 cm or less from the internal cervical opening. **Abbreviations:** BL/S, bilobed/succenturiate; IVF, in vitro fertilisation; LLP, low-lying placenta; TAS, transabdominal scan; TVS, transvaginal scan; VCI, velamentous cord insertion; VP, vasa praevia.

Critical Appraisal Worksheet for prognostic studies [32], the Quality Assessment of Diagnostic Accuracy Studies (QUADAS-2) tool for diagnostic studies [33], or the Drummond checklist for economic evaluations (S2 File) [34]. Literature-derived inputs were validated through expert opinion during two workshops involving six UK clinical experts (GA, BT, NT, AM, EDJ, HG) and two modelling experts (CH, ORA); these workshops were also used to inform inputs in the absence of published data. Workshop participants were selected based on their relevant roles and expertise within the UK NSC structures and/or involvement in previous UK NSC consultations.

Table 2 lists the key model inputs; a full list of all model inputs is provided in the S1 Table.

## Model outputs

In order to understand the potential impact of the different detection pathways, with regards to the outcomes in VP pregnancies as well as possible resource implications and trade-offs, a variety of outputs were modelled. These were the number of additional TAS scans and referrals

**Table 2. Key model inputs.**

| Input | Value | Rationale | Reference |
|---|---|---|---|
| **Overall model population** | | | |
| The total number of pregnancies in the UK population per year | 862,785 | Official UK statistics, providing accurate, population-level data | ONS [35] |
| **Proportion of women entering the VP screening pathway** | | | |
| No screening | 0.00% | Assumption that no women are tested as part of the no screening pathway | Assumed |
| VCI-based | 100.00% | Assumption that all pregnant women in the UK are initially tested for VCI (and BL/S placenta) as part of the VCI-based pathway | Assumed |
| IVF-based | 1.60% | Based on the prevalence of IVF-based pregnancies (see S1 Table) | Ebbing 2013 [36] |
| LLP-based | 10.00% | Based on the prevalence of LLP pregnancies (see S1 Table) | Expert opinion |
| **Incidence of VCI** | | | |
| General population | 1.50% | Appropriate literature value (based on applicability and quality of the study) identified through targeted searches | Ebbing 2013 [36] |
| IVF pregnancies | 3.70% | Appropriate literature value (based on applicability and quality of the study) identified through targeted searches | Ebbing 2013 [36] |
| LLP pregnancies | 2.80% | Appropriate literature value (based on applicability and quality of the study) identified through targeted searches | Suzuki 2015 [37] |
| **Incidence of VP** | | | |
| General population | 0.03% | Average value identified in the UK NSC review, in alignment with expert opinion | UK NSC 2017 [1] |
| IVF pregnancies | 0.34% | Appropriate literature value (based on applicability and quality of the study) identified through targeted searches | Schachter 2002 [38] |
| LLP pregnancies | 0.52% | Appropriate literature value (based on applicability and quality of the study) identified through targeted searches | Rosenberg 2011 [39] |
| **Diagnostic test accuracy** | | | |
| Sensitivity of TAS for VCI | 99.00% | Appropriate literature value (based on applicability and quality of the study) identified through targeted searches | Sepulveda 2003 [40] |
| Sensitivity of TAS for BL/S placenta | 75.00% (range: 65.00–85.00) | Appropriate literature value (based on applicability and quality of the study) identified through targeted searches | Cipriano 2010 [27] |
| Sensitivity of TAS for VP | 87.00% | Appropriate literature value (based on applicability and quality of the study) identified through targeted searches | Catanzarite 2001 [3] |
| Sensitivity of TVS for VP | 96.60% | Appropriate literature value (based on applicability and quality of the study) identified through targeted searches | Bronsteen 2013 [41] |

**Abbreviations:** BL/S, bilobed or succenturiate; IVF, in vitro fertilisation; LLP, low-lying placenta; TAS, transabdominal sonography; TVS, transvaginal sonography; UK NSC, United Kingdom National Screening Committee; VCI, velamentous cord insertion; VP, vasa praevia.

to TVS (as a direct outcome of the modelled pathways, i.e. not including referrals as part of current clinical practice), diagnosed and undiagnosed cases of VP, detected cases of VCI, the proportion of emergency Caesarean sections for VP pregnancies, and the number of VP-associated perinatal deaths. The ratio of additional TAS scans or referrals to TVS to the number of diagnosed VP cases was included as a simplified estimate of efficiency for each of the pathways.

The two outputs considered as main outcomes of interest for additional sensitivity analyses (see below for more details) were the proportion of diagnosed VP pregnancies, due to this being a key step towards the comparison of clinical outcomes, and number of referrals to TVS, as an important indicator of possible resource implications.

## Sensitivity and scenario analyses

In order to identify key drivers for the two main outcomes of interest in the model (diagnosed VP pregnancies and referrals to TVS), a deterministic sensitivity analysis (DSA) was

conducted to evaluate the impact of each model parameter on the difference in these outcomes between each pathway and no screening. Where possible, published variance data (e.g. confidence intervals [CI]) were used to perform the DSA; in the absence of published data, approximate 95% CIs were calculated from the base case values using the Wilson score interval method for binomial probabilities [42].

A probabilistic sensitivity analysis (PSA) was carried out to evaluate the joint parameter uncertainty across model inputs on the two main outcomes of interest. Input values were varied stochastically based on published variance data where possible, or calculated Wilson score intervals where required in the absence of published data, and using beta distributions (as all included inputs were binomial probabilities) [43]. Each new combination of input values was tested in turn during 1,000 iterative simulations, and a plot was generated showing the mean average difference (and associated non-parametric 95% CI) between each pathway and no screening with regards to the number of referrals to TVS and diagnosed VP pregnancies.

Scenario analyses were run to explore both uncertainties associated with model inputs and structural assumptions concerning the detection pathways. To explore uncertainty in the model inputs, a set of alternative literature values for key model inputs (Alternative Inputs Scenario) were used simultaneously. The key model inputs were informed by the inputs that were found to have the greatest impact on the proportion of diagnosed VP in the DSA (these inputs, and the alternative values used, are summarised in S2 Table). To account for the assumption that accuracy of diagnostic testing for VP may improve over time, based on likely evolving technology and clinical practice, an additional scenario analysis was based on the inclusion of higher sensitivity inputs for TVS for VP (100%; based on Ruiter et al.) [44] and for TAS for VP (98%; assumed to be slightly lower than TVS).

To explore structural assumptions, two Structural Scenarios were developed. In Structural Scenario 1, the TAS for VP step at 32 weeks was removed from all pathways. Structural Scenario 2 included a combined IVF- and LLP-based pathway in which pregnancies resulting from IVF and/or with LLP were considered eligible for TAS at 32 weeks. For this scenario, published odds ratios for the occurrence of LLP in IVF pregnancies were used to calculate the combined IVF/LLP cohort, and VCI and VP incidence values in either IVF (for VCI) or LLP (for VP) pregnancies were conservatively assumed for this cohort. All scenario analyses were decided through discussion with clinical experts during two independent workshops, taking into account alternative scenarios that were deemed plausible in clinical practice and of the most interest for this exploratory analysis.

## Results

### Base case

In the no screening pathway, no pregnancies were actively tested for VP. However, 0.003% (27 pregnancies) of all pregnancies were incidentally diagnosed as VP at $18^{+0}$ to $20^{+6}$ weeks and directly referred to confirmatory TVS. 14,126 pregnancies in the IVF-based pathway underwent TAS specifically for VCI and BL/S placenta at $18^{+0}$ to $20^{+6}$ weeks; for the VCI-based pathway, all of the 862,785 pregnancies that occurred within the modelled one-year UK pregnancy cohort underwent TAS for VCI and BL/S placenta in addition to the routine fetal anomaly scan at $18^{+0}$ to $20^{+6}$ weeks. In the LLP-based pathway 10.0% (86,270 pregnancies) of all pregnancies had LLP, with 85,407 (99.0%) of these being detected as part of the routine fetal anomaly scan at $18^{+0}$ to $20^{+6}$ weeks and referred to follow-up examinations, with TAS for VP added to current practice, at 32 weeks accordingly (Table 3).

Correspondingly, the LLP-based pathway resulted in the highest number of referrals to 32-week TAS and subsequent referrals to TVS (85,407 and 2,083 referrals, respectively),

**Table 3. Base case results for demographic and screening outcomes.**

| | No screening | IVF-based pathway | LLP-based pathway | VCI-based pathway |
|---|---|---|---|---|
| **VP pregnancies, n [difference vs no screening pathway]** | | | | |
| Within the affected population entering the pathway | 0 | 62 | 448 | 700 |
| | | [+62] | [+448] | [+700] |
| Within the affected population *not* entering the pathway | 700 | 638 | 252 | 0 |
| | | [−62] | [−448] | [−700] |
| Total | 700 | 700 | 700 | 700 |
| | | [0] | [0] | [0] |
| **VCI pregnancies, n [difference vs no screening pathway]** | | | | |
| Within the affected population entering the pathway | 0 | 447 | 2,397 | 14,361 |
| | | [+447] | [+2,397] | [+14,361] |
| Within the affected population *not* entering the pathway | 14,361 | 13,914 | 11,964 | 0 |
| | | [−447] | [−2,397] | [−14,361] |
| Total | 14,361 | 14,361 | 14,361 | 14,361 |
| | | [0] | [0] | [0] |
| **Number of scans, n [difference vs no screening pathway]** | | | | |
| Additional TAS scans at $18^{+0}$ to $20^{+6}$ weeks [a] | 0 | 14,126 | 0 | 862,785 |
| | | [+14,126] | | [+862,785] |
| Referrals to 32-week TAS | 0 | 834 | 85,407 | 38,028 |
| | | [+834] | [+85,407] | [+38,028] |
| TVS scans for VP | 27 | 66 | 2,083 | 1,319 |
| | | [+39] | [+2,056] | [+1,292] |
| **Screening outcomes** | | | | |
| VCI detected, n (% of all VCI pregnancies) [difference vs no screening pathway] | 0 (0) | 443 (3.1) | 0 (0) | 14,238 (99.1) |
| | | [+443 (+3.1)] | [0] | [+14,238 (+99.1)] |
| VP diagnosed, n (% of all VP pregnancies) [difference vs no screening pathway] | 27 (3.9) | 50 (7.1) | 371 (53.5) | 552 (78.9) |
| | | [+23 (+3.2)] | [+344 (+49.6)] | [+525 (+75.0)] |
| TAS scans per VP diagnosed, n [difference vs no screening pathway] [b] | 0 | 299 | 230 | 1,632 |
| | | [+299] | [+230] | [+1,631] |
| TVS scans per VP diagnosed, n [difference vs no screening pathway] | 1.0 | 1.3 | 5.6 | 2.4 |
| | | [+0.3] | [+4.6] | [+1.4] |

[a] TAS scans (for VCI) performed in addition to the routine fetal anomaly TAS at 18+0 to 20+6 weeks b Including additional TAS for VCI at 18+0 to 20+6 weeks (for the IVF- and VCI-base pathways) and TAS for VP at 32 weeks (for the IVF-, LLP- and VCI-based pathways)

**Abbreviations:** IVF, in vitro fertilisation; LLP, low-lying placenta; TAS, transabdominal sonography; TVS, transvaginal sonography; VCI, velamentous cord insertion; VP, vasa praevia.

followed by VCI-based screening (38,028 and 1,319 referrals, respectively). While a higher number of referrals may appear counterintuitive for a more targeted screening approach, this is in keeping with the higher incidence of LLP in the general population (10.0%) compared to VCI (1.5%) and BL/S placenta (3.1%) (see S1 Table), and should also be regarded in the overall context of fewer additional TAS scans being performed at $18^{+0}$ to $20^{+6}$ weeks as part of this targeted approach when compared with VCI-based screening (Table 3). The IVF-based pathway resulted in substantially fewer TAS and TVS referrals (834 and 66 respectively), and only 27 referrals to TVS due to incidental detection occurred in the no screening pathway (Table 3). At 32 weeks, the rate of false positive VP diagnoses was very low in all pathways, with two false positives in the LLP-based pathway and one false positive in the VCI-based pathway. VCI-based screening diagnosed the highest proportion of VP pregnancies (78.9%, 552 pregnancies)

followed by the LLP-based pathway (53.5%, 371 pregnancies). 50 VP pregnancies (7.0%) were identified in the IVF-based pathway, and the no screening pathway led to the incidental detection of only 27 VP pregnancies (3.9%; Table 3). When considering a simplified measure of efficiency, the LLP-based pathway resulted in the lowest number of additional TAS scans (for VCI or VP) for each diagnosed case of VP (230 TAS scans) but also the highest number of TVS scans per diagnosed VP (5.6 TVS scans), when compared with the IVF-based (299 TAS and 1.3 TVS scans per diagnosed VP) and VCI-based (1,632 TAS and 2.4 TVS scans per diagnosed VP) pathways (Table 3). These results should however also be regarded in the context of the actual detection algorithms and rates for each pathway, as exemplified by the no screening pathway resulting in a seemingly perfect ratio of 1.0 TVS scan per diagnosed case of VP based on the incidental detection (and direct referral to confirmatory TVS) of 27 VP pregnancies during the $18^{+0}$ to $20^{+6}$ weeks routine scan.

Despite the overall incidence of VCI being equal across all four pathways, it was assumed that no VCI pregnancies were detected in the LLP-based or no screening pathways, due to VCI not being actively tested for or considered as part of the respective detection strategies in these two pathways (Table 3). 443 VCI pregnancies (3.1%) were detected in the IVF-based pathway, while 14,238 VCI pregnancies (99.1%) were detected under VCI-based screening. For the VCI-based pathway, it should also be noted that this number is considerably smaller than the 38,028 pregnancies that were referred to 32-week TAS for VP due to the false positive detection of VCI at $18^{+0}$ to $20^{+6}$ weeks.

The pathways with higher proportions of diagnosed VP cases also resulted in a higher proportion of VP pregnancies being delivered via planned Caesarean section (Table 4). In the VCI-based and LLP-based pathways, 61.6% and 42.6% of all VP pregnancies were delivered via planned Caesarean section, respectively, compared with 9.0% and 6.6% of all VP pregnancies in the IVF-based and no screening pathways. A higher proportion of VP pregnancies were therefore delivered via emergency Caesarean section (64.9% and 66.3% of all VP pregnancies, respectively) or vaginal births (26.1% and 27.1%) in the IVF-based and no screening pathways, compared to the VCI-based and LLP-based pathways (emergency Caesarean section: 32.5% and 44.1%; vaginal delivery: 5.9% and 13.3%, respectively). In line with the increased proportion of planned Caesarean sections versus vaginal births for VP pregnancies in the VCI-based and LLP-based pathways, the proportion of VP pregnancies resulting in perinatal death was substantially lower in these pathways compared to the no screening and IVF-based pathways (Table 4). The VCI-based pathway resulted in 100 perinatal deaths in VP pregnancies (14.2% of all VP pregnancies), and the LLP-based pathway resulted in 196 perinatal deaths (28.0%).

**Table 4. Base case results for birth method and perinatal outcomes.**

| | No screening | IVF-based pathway | LLP-based pathway | VCI-based pathway |
|---|---|---|---|---|
| Planned Caesarean Sections for VP pregnancies, n (% of all VP pregnancies) [difference vs no screening pathway] | 46 (6.6) | 63 (9.0) | 298 (42.6) | 431 (61.6) |
| | | [+17 (+2.4)] | [+252 (36.0)] | [+385 (+55)] |
| Emergency Caesarean Sections for VP pregnancies, n (% of all VP pregnancies) [difference vs no screening pathway] | 464 (66.3) | 454 (64.9) | 309 (44.1) | 227 (32.5) |
| | | [-10 (-1.4)] | [-155 (-22.2)] | [-237 (-33.8)] |
| Vaginal deliveries for VP pregnancies, (% of all VP pregnancies) [difference vs no screening pathway] | 190 (27.1) | 183 (26.1) | 93 (13.3) | 42 (5.9) |
| | | [-7 (-1.0)] | [-97 (-13.8)] | [-148 (21.2)] |
| Perinatal deaths in VP pregnancies, n (% of VP pregnancies) [difference vs no screening pathway] | 380 (54.3) | 368 (52.6) | 196 (28.0) | 100 (14.2) |
| | | [-12 (-1.7)] | [-184 (-26.3)] | [280 (-40.0)] |

**Abbreviations:** IVF, in vitro fertilisation; LLP, low-lying placenta; VCI, velamentous cord insertion; VP, vasa praevia.

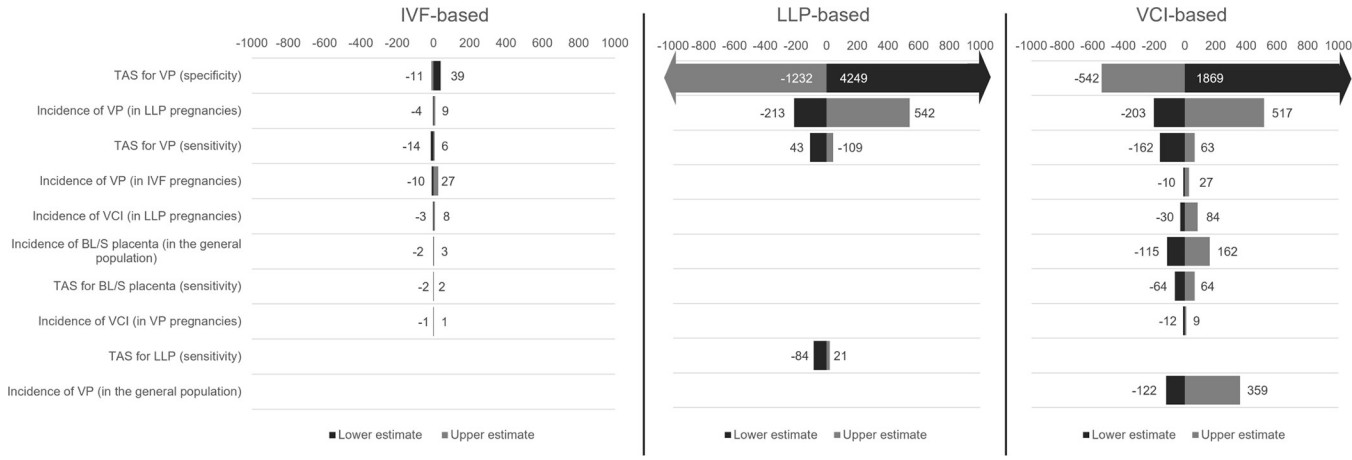

**Fig 2. Results of the DSA for the number of referrals to TVS.** Upper estimate demonstrates the impact on the difference between each pathway and the no screening pathway in terms of referrals to TVS by increasing the variable; lower estimate demonstrates the impact on the referrals to TVS by decreasing the variable. Asymmetric bars are indicative of input values already being close to the ceiling value for the input type (for example, a probability of 0.9) and therefore being unable to be increased to the full extent. **Abbreviations**: BL/S, bilobed or succenturiate; DSA, deterministic sensitivity analysis; IVF, in vitro fertilisation; LLP, low-lying placenta; TAS, transabdominal sonography; TVS, transvaginal sonography; VCI, velamentous cord insertion; VP, vasa praevia.

Meanwhile, the IVF-based pathway resulted in 368 deaths (52.6%) and no screening resulted in 380 deaths (54.3%).

## Sensitivity analyses

The results of the DSA display the impact of the key model drivers on the difference between no screening and each of the IVF-based, LLP-based and VCI-based detection pathways for the number of referrals to TVS (Fig 2) and diagnosed VP cases (Fig 3). For both of these outcomes, the results of the PSA further demonstrate that joint parameter uncertainty across model inputs led to some variation in the difference between no screening and the LLP-based and

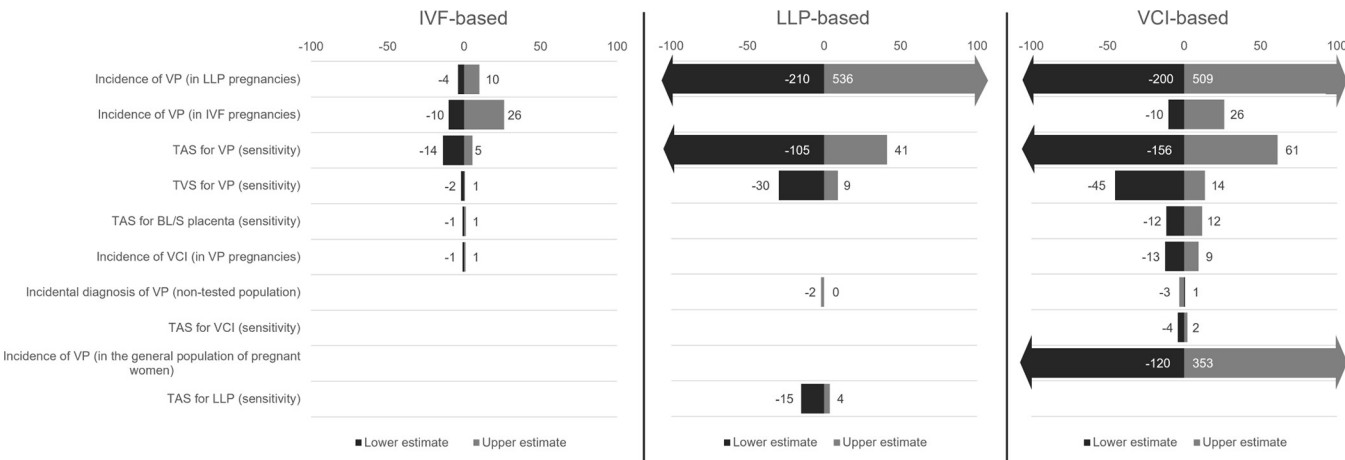

**Fig 3. Results of the DSA for the number of diagnosed VP cases.** Upper estimate demonstrates the impact on the difference between each pathway and the no screening pathway in terms of diagnosed VP pregnancies by increasing the variable; lower estimate demonstrates the impact on the number of diagnosed VP pregnancies by decreasing the variable. Asymmetric bars are indicative of input values already being close to the ceiling value for the input type (for example, a probability of 0.9) and therefore being unable to be increased to the full extent. **Abbreviations:** BL/S, bilobed or succenturiate; DSA, deterministic sensitivity analysis; IVF, in vitro fertilisation; LLP, low-lying placenta; TAS, transabdominal sonography; TVS, transvaginal sonography; VCI, velamentous cord insertion; VP, vasa praevia.

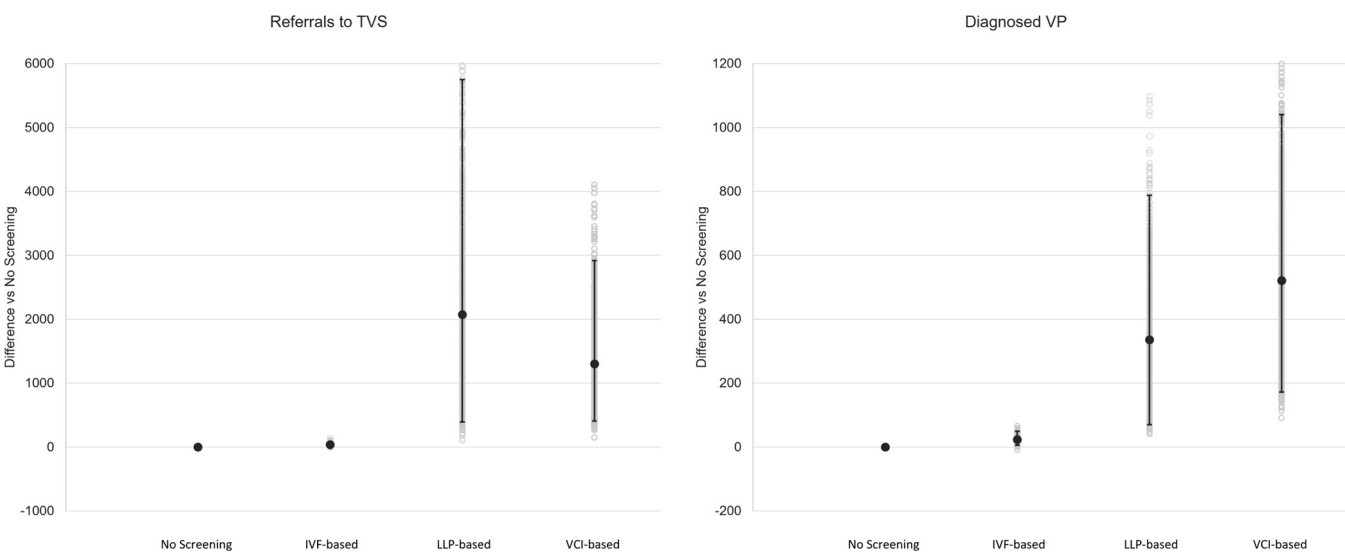

**Fig 4. Results of the PSA for the difference versus no screening (number of referrals to TVS and number of diagnosed VP cases).** Upper estimate demonstrates the impact on the difference between each pathway and the no screening pathway in terms of diagnosed VP pregnancies by increasing the variable; lower estimate demonstrates the impact on the number of diagnosed VP pregnancies by decreasing the variable. Asymmetric bars are indicative of input values already being close to the ceiling value for the input type (for example, a probability of 0.9) and therefore being unable to be increased to the full extent. **Abbreviations**: BL/S, bilobed or succenturiate; DSA, deterministic sensitivity analysis; IVF, in vitro fertilisation; LLP, low-lying placenta; TAS, transabdominal sonography; TVS, transvaginal sonography; VCI, velamentous cord insertion; VP, vasa praevia.

VCI-based pathways in particular (Fig 4). Here, a noticeable overlap between these two pathways was observed with regards to the mean average difference versus no screening for referrals to TVS (LLP-based: 2,075 [95% CI: 391, 5,750]; VCI-based: 1,301 [95% CI: 409, 2,920]) and the number of diagnosed VP cases (LLP-based: 336 [95% CI: 70, 788]; VCI-based: 521 [95% CI: 172, 1,041]).

## Scenario analyses

The results of all scenario analyses are presented in Fig 5. The simultaneous incorporation of alternative literature values for ten key inputs in the Alternative Inputs Scenario resulted in a substantial decrease in both the proportion of diagnosed VP and detected VCI pregnancies in the VCI-based pathway, accompanied by a corresponding increase in the proportion of perinatal death in VP pregnancies; this was also observed, to a lesser degree, for the LLP-based pathway. A substantial increase in the proportion of detected VCI pregnancies was also observed in the IVF-based pathway. The proportion of diagnosed VP and perinatal death in VP pregnancies in the no screening pathway, as well as the number of referrals to TVS in all four pathways, remained comparatively unchanged.

The increase of test sensitivity for TAS and TVS for VP resulted in a substantially increased proportion of diagnosed VP cases for the LLP-based (62%) and VCI-based (92%) pathways in particular, with correspondingly lower proportions of perinatal death in VP pregnancies (23% and 7% in the LLP-based and VCI-based pathway, respectively).

Removal of TAS for VP at 32 weeks in Structural Scenario 1 resulted in a substantially increased number of referrals to TVS in the LLP-based, VCI-based, and IVF-based pathways, with correspondingly increased rates of VP detection in the VCI-based and LLP-based pathways, in particular. In Structural Scenario 2, combining the IVF- and LLP-based pathways generated very similar results to the LLP-based pathway.

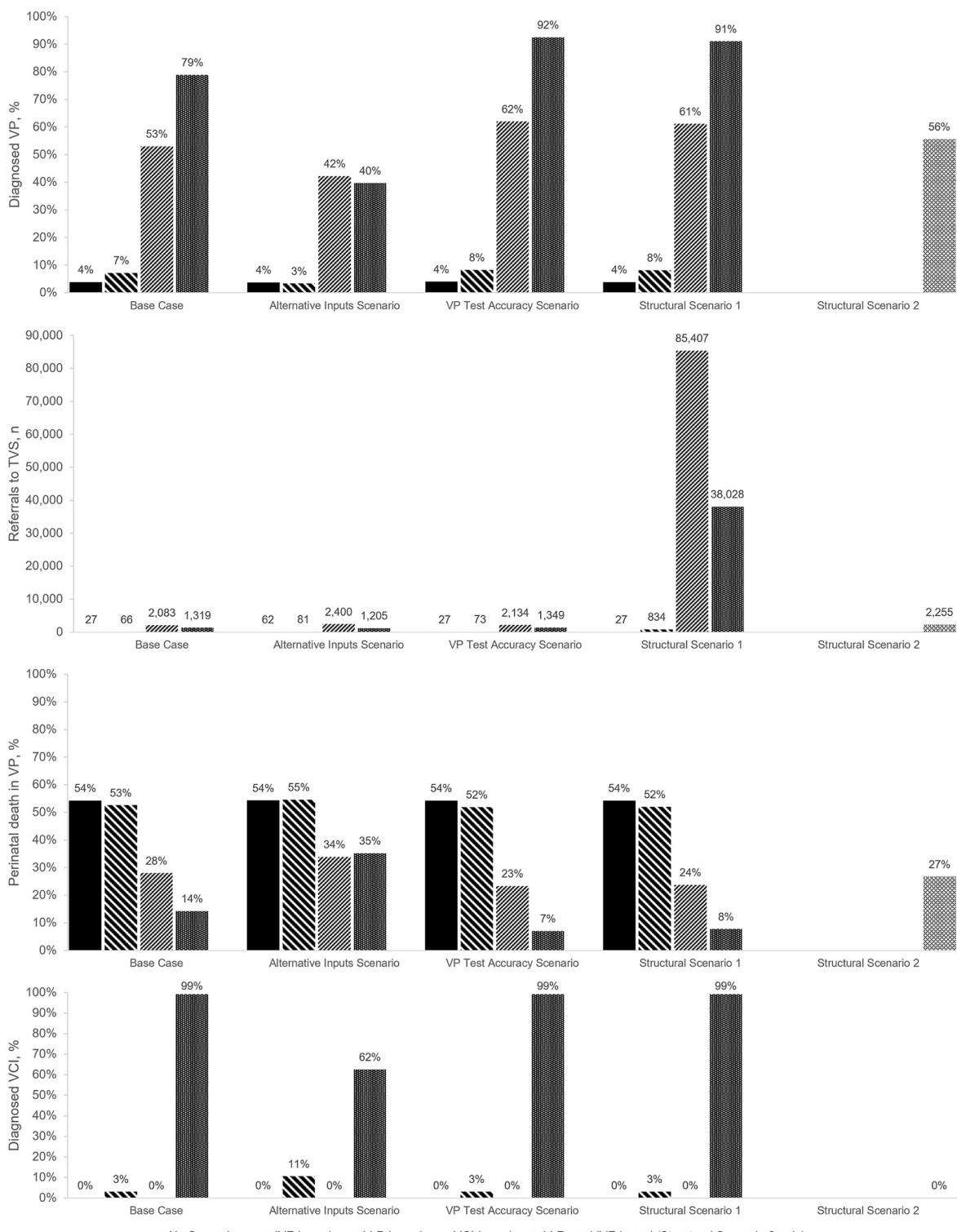

**Fig 5. Results of scenario analyses. Alternative Inputs Scenario:** Incorporation of alternative inputs based on alternate literature values. **VP Test Accuracy Scenario:** Higher test sensitivity for TAS and TVS for VP. **Structural Scenario 1:** Removal of TAS for VP at 32 weeks. **Structural Scenario 2**: Combined IVF- and LLP-based pathway. **Abbreviations:** BL/S, bilobed or succenturiate; IVF, in vitro fertilisation; LLP, low-lying placenta; TAS, transabdominal sonography; TVS, transvaginal sonography; UKOSS, UK Obstetric Surveillance System; VCI, velamentous cord insertion; VP, vasa praevia.

## Discussion

The base case results of this exploratory study showed that the modelled VCI-based and LLP-based pathways led to the detection of a greater proportion of VP pregnancies and a higher number of referrals to TVS than the no screening or IVF-based pathways. These higher VP detection rates also led to a correspondingly lower proportion of VP pregnancies resulting in perinatal death. The VCI-based pathway resulted in the highest VP detection rate (78.9%) and lowest proportion of perinatal death in VP pregnancies (14.2%); however, it also resulted in the detection of almost all VCI pregnancies, compared to minimal detection of VCI in the other pathways, and required a substantially higher number of additional TAS scans which, currently, are rarely recommended in practice. In contrast, the LLP-based pathway diagnosed a lower proportion of VP (53.5%) but required significantly fewer additional TAS scans; this pathway also did not include the detection of VCI as part of its screening algorithm.

The major limitation of this modelling study is the considerable uncertainty associated with many of the model inputs due to the lack of consistent high-quality data. The evidence base available for prevalence estimates of VP and corresponding risk factors, as well as diagnostic test accuracy, is generally characterised by a high degree of heterogeneity and mixed quality of reporting [4, 14, 44]. Formal quality appraisals also indicated that available studies used to inform model parameters were generally of low or moderate quality. This applied to key parameters such as test accuracy and the incidence and impact of both VP and VCI. Therefore, an Alternative Inputs Scenario analysis was used to explore this uncertainty by applying alternative literature-derived inputs. This resulted in a pronounced decrease in the detection of VP in the VCI-based pathway below the rate of detection in the LLP-based pathway, likely driven by the considerably lower scenario input for VCI test sensitivity. This finding is further supported by the results of the PSA, which demonstrated a noticeable overlap in the number of referrals to TVS and detected VP cases for the VCI-based and LLP-based pathways. Collectively, the results of the sensitivity and scenario analyses indicate that the relative benefits of individual pathways, and especially VCI-based screening, remain uncertain to some degree and the general lack of high-quality evidence should prompt caution when interpreting the results of this exploratory analysis.

Reduction of VP-related mortality is directly linked to the VP detection rate and is therefore also impacted by the uncertainty of this latter outcome, with this also being supported by the results of a scenario analysis which applied a higher VP diagnostic test sensitivity and resulted in noticeably lower numbers of perinatal deaths in VP pregnancies. Similarly, while an intermediate measure of screening efficiency was based on VP detection as the more immediate key outcome in the model, the observed trends with regards to the number needed to screen for detecting VP in each pathway would also apply to VP-related deaths prevented, which would be the ultimate aim of any VP screening strategy.

Additional uncertainty surrounds the mortality associated with ultrasound-detected VP compared to clinically presenting VP at birth. The model base case estimated that 380 (54.3%) of 700 cases of VP would result in perinatal death without antenatal ultrasound screening. This case-fatality rate is informed by the literature on clinically presenting VP and is also aligned with the conclusions made from a UK single-centre study by Zhang et al. where the authors estimated that around half of the 21 ultrasound-detected cases of VP (in a cohort of 26,830 pregnancies) would have resulted in stillbirth if they had not been diagnosed prenatally [2, 20]. However, these estimates of VP-related mortality contrast with a 2017 national clinical surveillance study conducted in the UK where, in a cohort of approximately 750,000 pregnancies in a setting in which screening for VP is not recommended in national guidance, six deaths to VP were reported as part of the currently available preliminary results [19, 45]. Whilst the

potential impact of under-reporting in this surveillance study may require further consideration, any large difference between the assumed and observed number of perinatal deaths may also be explained by the different diagnostic criteria for ultrasound-detected VP, with Zhang et al. having applied a definition based on vessels within 5 cm of the internal os as diagnostic criterion for VP [20]. This broadened diagnostic definition of VP may not correlate with VP which presents clinically in labour and a lower mortality per case detected might therefore be expected in screen detected VP. This would also impact any estimates of the number needed to screen to prevent VP-related mortality. When estimating the impact of antenatal detection strategies on VP-related mortality, caution should therefore be exercised when extrapolating outcomes from clinically presenting VP to ultrasound-detected VP.

This exploratory model is also consistent with two other, Canadian and US-based, modelling studies which investigate the cost-effectiveness of VP screening [27, 28]. Although the exploratory model presented here does not consider cost-effectiveness, the results align with these published studies with respect to the proportion of diagnosed VP under the VCI-based (as approximate to population screening) and LLP-based pathways.

One finding of the model was that the modelled LLP-based pathway resulted in a substantially higher number of referrals to TAS at 32 weeks compared to the VCI-based pathway. This should, however, also be interpreted in the context of current clinical practice. As LLP is routinely detected at the $18^{+0}$ to $20^{+6}$ week scan and women with LLP are usually re-scanned at 32 weeks [13], this already existing rate of LLP-related referrals would thus be present in any potential detection strategy in practice. Therefore, the LLP-based pathway would require only minimal additional TAS resource overall specifically for the detection of VP compared to current practice. In contrast, the VCI-based pathway would require additional screening for VCI and BL/S placenta in all pregnancies undergoing the routine fetal anomaly scan at $18^{+0}$ to $20^{+6}$ weeks. It should however be noted that this may overestimate the number of additionally required scans, as reporting of placental cord insertion during the $18^{+0}$ to $20^{+6}$ week scan may already be routinely practiced in some UK centres. Irrespective, the referral of all pregnancies with detected VCI would always result in the additional recall of large numbers of women (38,028 in the base case) with positive screening results to TAS for VP at 32 weeks and subsequent referral to TVS where indicated.

Crucially, a high proportion of the VCI pregnancies detected as part of this pathway would not be affected by VP and a substantial number of VCI pregnancies diagnosed at $18^{+0}$ to $20^{+6}$ weeks would actually be false positives (around 24,000 in the base case). The situation relating to VCI is therefore complicated, and it has also been noted that VCI, and cord anomalies more generally, currently represent an area of obstetrics which has not been well studied [46]. At the same time, some evidence points towards a small absolute increase in the risk of adverse pregnancy outcomes for VCI pregnancies [1, 46, 47], but no evidence-based interventions or management pathways are available to reduce such risks. Overdetection and false positive test results may cause unnecessary anxiety and, with a limited evidence base it may be challenging to develop high quality information to mitigate this. So while the association between VP and VCI has led to some guidelines recommending that all women are screened for VCI [9, 10], this would be a departure from current UK practice and the uncertainties relating to key elements of a screening and management pathway for such cord anomalies have been highlighted [46]. As such, there is uncertainty about the balance of clinical benefit and harm that may result from screen detection of VCI, particularly in the absence of VP. In contrast, adding the offer of testing for VP alongside already performed scans for placenta praevia in late pregnancy in a limited number of women may represent a more targeted approach compared to a VCI-based pathway in areas, like the UK, where there is no nationally recommended strategy for detection of VP or cord anomalies.

However, given the exploratory nature of the reported analyses and the considerable uncertainty associated with many of the model inputs, further investigation is required regarding the potential effect of the different detection strategies and there is a need for high-quality data to inform discussions about VP screening in the UK.

In conclusion, the results of this modelling exercise suggest that a targeted LLP-based approach could detect a substantial proportion of VP cases while avoiding the potential complications from the detection of VCI and requiring minimal changes to current clinical practice. However, without further research, future discussions about screening for VP will continue to be constrained by the lack of high-quality data encountered in this exploratory study.

## Supporting information

**S1 Fig. Structure of the VP screening model.**
(TIF)

**S1 Table. Full list of inputs, including rationale and references.**
(DOCX)

**S2 Table. Alternative inputs used in the Alternative Inputs scenario analysis.**
(DOCX)

**S1 File. SLR and MA on adverse perinatal outcomes.**
(DOCX)

**S2 File. Quality assessment results.**
(DOCX)

**S3 File. Glossary.**
(DOCX)

**S4 File. Model spreadsheet.**
(XLSX)

## Acknowledgments

The authors acknowledge Elizabeth Daly-Jones, Lead Sonographer at Imperial NHS Trust, and Hilary Goodman, Operational Manager–Antenatal Services/Screening at Hampshire Hospitals NHS Foundation Trust, for substantial contributions to study design. The authors also acknowledge Helen Bewicke-Copley MSc, Kate Hanman MSc, Hattie Cant BSc, Ania Bobrowska PhD and Annabel Griffiths PhD from Costello Medical, UK, for medical writing and editorial assistance in preparing this manuscript for publication, based on the authors' input and direction.

## Author Contributions

**Conceptualization:** Benjamin Ruban-Fell, George Attilakos, Tao Haskins-Coulter, Christopher Hyde, Jeanette Kusel, Anne Mackie, Oliver Rivero-Arias, Basky Thilaganathan, Nigel Thomson, Cristina Visintin, John Marshall.

**Formal analysis:** Benjamin Ruban-Fell.

**Methodology:** Benjamin Ruban-Fell, George Attilakos, Tao Haskins-Coulter, Christopher Hyde, Jeanette Kusel, Anne Mackie, Oliver Rivero-Arias, Basky Thilaganathan, Nigel Thomson, Cristina Visintin, John Marshall.

**Validation:** Benjamin Ruban-Fell, George Attilakos, Tao Haskins-Coulter, Christopher Hyde, Jeanette Kusel, Anne Mackie, Oliver Rivero-Arias, Basky Thilaganathan, Nigel Thomson, Cristina Visintin, John Marshall.

**Writing – original draft:** Benjamin Ruban-Fell, John Marshall.

**Writing – review & editing:** Benjamin Ruban-Fell, George Attilakos, Tao Haskins-Coulter, Christopher Hyde, Jeanette Kusel, Anne Mackie, Oliver Rivero-Arias, Basky Thilaganathan, Nigel Thomson, Cristina Visintin, John Marshall.

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
