## [Decision Letter · Decision Letter 0]

7 Jul 2022

PONE-D-21-26780

The impact of ultrasound-based antenatal screening strategies to detect vasa praevia in the United Kingdom: an exploratory study using decision analytic modelling methods

PLOS ONE

Dear Dr. Ruban-Fell,

Thank you for submitting your manuscript to PLOS ONE. After careful consideration, we feel that it has merit but does not fully meet PLOS ONE’s publication criteria as it currently stands. Therefore, we invite you to submit a revised version of the manuscript that addresses the points raised during the review process.

Please note that we have only been able to secure a single reviewer to assess your manuscript. We are issuing a decision on your manuscript at this point to prevent further delays in the evaluation of your manuscript. Please be aware that the editor who handles your revised manuscript might find it necessary to invite additional reviewers to assess this work once the revised manuscript is submitted. However, we will aim to proceed on the basis of this single review if possible. 

We look forward to receiving your revised manuscript.

Kind regards,

Vanessa Carels

Staff Editor

PLOS ONE

Journal Requirements:

2. Please expand the acronym “UK NSC” (as indicated in your financial disclosure) so that it states the name of your funders in full.

3. Thank you for stating the following in the Competing Interests section: "All authors have completed the ICJME uniform disclosure form at www.icmje.org/coi_disclosure.pdf (available on request from the corresponding author) and declare: no support from any organisation for the submitted work other than that described above; JM, CV, and AM are employees of the UK NSC secretariat which funded the submitted work; CH is a member of the UK NSC; BT and ORA are members of the Fetal, Maternal and Child Health Group (FMCH) of the UK NSC; GA is a Council member of the Royal College of Obstetricians and Gynaecologists and a Steering Committee member of the UK Obstetric Surveillance System; BRF, JK and THC are, or were formerly, employed by Costello Medical which was commissioned for the model development and supportive work by the UK NSC; no other relationships or activities that could appear to have influenced the submitted work."

Reviewers' comments:

Reviewer's Responses to Questions

**Comments to the Author**

1. Is the manuscript technically sound, and do the data support the conclusions?

Reviewer #1: Yes

2. Has the statistical analysis been performed appropriately and rigorously? 

Reviewer #1: Yes

3. Have the authors made all data underlying the findings in their manuscript fully available?

Reviewer #1: Yes

4. Is the manuscript presented in an intelligible fashion and written in standard English?

Reviewer #1: Yes

5. Review Comments to the Author

Reviewer #1: The authors carried out an exploratory study using decision analytic modelling methods examining various targeted screening methods for vasa previa in the United Kingdom. They evaluate screening for vasa previa in a hypothetical UK population in patients with low-lying placentas, velamentous cord insertions, and pregnancies resulting from IVF, and also with no screening.

They find that examining VCI based screening and low-lying placentas resulted in the highest detection rates and lowest rates of perinatal death.

The manuscript is well-written and the methodology is sound. The conclusions are supported by the methods. However, any model is only as good as the assumptions put into it.

Perhaps a flaw of this study, and of several others is the question: “What constitutes “screening” for vasa previa?” I suggest that the authors define clearly what is meant by “screening” for vasa previa. A flaw of this study and of the UK screening evaluations at present is that it makes the assumption that screening involves transvaginal ultrasound with Doppler. This is more diagnosis, and the authors continue perpetuating the myth that vasa previa cannot be routinely screened for (which could be done via routinely identifying placental cord insertion and a Doppler sweep of the lower uterine segment.

How do THESE authors define “screening” for vasa previa? Perhaps the only thing I find in their manuscript is “recall at 32 weeks to perform further TAS to confirm the presence of VP (representing a population screening strategy based on a risk factor not currently reported in UK practice)” (Line 118)

The authors also do not examine the role of universal screening as part of their study. Wouldn’t it be worth seeing how universal screening performed in this context? Perhaps tied into this is the question above: “What constitutes screening for vasa previa?

The authors state in line 122 “there is no independently significant association between multiple pregnancies and VP incidence.” Is this actually true?

The UK NSC estimate of the incidence of VP of 0.03% of pregnancies is likely an underestimation. The authors have addressed this. However, the UKOSS may have its own inherent flaws.

An important finding is that clearly using IVF alone as the screening for vasa previa will detect few cases, and lead to the highest number of deaths from VP.

6. PLOS authors have the option to publish the peer review history of their article (what does this mean?). If published, this will include your full peer review and any attached files.

Reviewer #1: No

---

## [Author Response · Author response to Decision Letter 0]

1 Sep 2022

Reviewer 1

Question/Comment: The authors carried out an exploratory study using decision analytic modelling methods examining various targeted screening methods for vasa previa in the United Kingdom. They evaluate screening for vasa previa in a hypothetical UK population in patients with low-lying placentas, velamentous cord insertions, and pregnancies resulting from IVF, and also with no screening. They find that examining VCI based screening and low-lying placentas resulted in the highest detection rates and lowest rates of perinatal death.

The manuscript is well-written and the methodology is sound. The conclusions are supported by the methods. However, any model is only as good as the assumptions put into it..

Response: We thank the reviewer for this comment and the positive assessment of the presented manuscript. We agree with the raised point that the reliability of a model is largely dependent on the reliability of the underlying assumptions. One of the reasons we undertook this study was to draw attention to the need for research in this important area, and we hope that the inclusion of a full set of sensitivity and relevant scenario analyses as part of the presented results will also go towards accounting for the reliance on assumptions and limited available data. 

Already with regards to characterisation of the different modelled strategies (which are also discussed in more detail further below), the reviewer might note that one of the strategies (velamentous cord insertion (VCI)) has been designed to represent an universal, or population, screening strategy and another (IVF-based) strategy has been described as a representation of risk assessment in routine clinical care.

Question/Comment: Perhaps a flaw of this study, and of several others is the question: “What constitutes “screening” for vasa previa?” I suggest that the authors define clearly what is meant by “screening” for vasa previa. A flaw of this study and of the UK screening evaluations at present is that it makes the assumption that screening involves transvaginal ultrasound with Doppler. This is more diagnosis, and the authors continue perpetuating the myth that vasa previa cannot be routinely screened for (which could be done via routinely identifying placental cord insertion and a Doppler sweep of the lower uterine segment.

How do THESE authors define “screening” for vasa previa? Perhaps the only thing I find in their manuscript is “recall at 32 weeks to perform further TAS to confirm the presence of VP (representing a population screening strategy based on a risk factor not currently reported in UK practice)” (Line 118).

Response: Many thanks for this comment. We have tried to address the request to state an overall definition of screening by adding this to the relevant paragraph in the Introduction. We have also included further argumentation (both within the Introduction as well as the description of the different strategies within the Methods section) to highlighted the reasoning behind drawing a distinction between the VCI strategy, as an example of universal/population screening, and the LLP strategy as an example of targeted screening (please note that we have also included an additional reference for this term/concept in the form of a recently published review by Bobrowska et al. 2022). 

Interestingly, available guidance by the Royal Australian and New Zealand College of Obstetricians and Gynaecologists (RANZCOG) also refers to a universal screening strategy based on the detection of VCI (followed by confirmation including TVS); we take this as some confirmation of our use of the term. 

We acknowledge that, while concepts and categories aim to assist orientation to problems, they are always also open to debate. We hope that the amendments we have provided will help the reviewer, as well as the eventual reader, to reflect on the conceptual differences between the strategies.

Regarding the point about transvaginal sonography (TVS), we would like to emphasise that we agree with the reviewer that TVS is a diagnostic procedure in this context and that abdominal ultrasound would constitute the screening procedure. However, it is important to consider the throughput from screening to diagnostic services and one of the aims of the paper is to also explore this aspect of the different strategies described in the model. We would also like to refer again to the available RANZCOG guidance as another example, outside of the UK context, that considers a confirmatory step (including TVS) as part of their recommended universal screening strategy, providing additional support for the inclusion of this step as part of our modelled strategy (which was initially informed and validated through consultation with a selection of relevant UK experts).

Question/Comment: The authors also do not examine the role of universal screening as part of their study. Wouldn’t it be worth seeing how universal screening performed in this context? Perhaps tied into this is the question above: “What constitutes screening for vasa previa?

Response: Many thanks for this comment. We hope that we have addressed this concern as part of the above response and the corresponding amendments to the manuscript. 

In keeping with the RANZCOG guideline, we consider the VCI-based strategy to be a suitable universal, or population, screening strategy, whilst also recognising that this may be open to debate.

Question/Comment: The authors state in line 122 “there is no independently significant association between multiple pregnancies and VP incidence.” Is this actually true?

Response: Many thanks for this query. The statement in question is informed by a thorough literature review commissioned by the UK NSC, which is further corroborated by the findings of a widely cited systematic literature review/meta-analysis performed by Ruiter et al. 2016 (now also added as supporting reference for this sentence in the manuscript).

However, we acknowledge that both publications further state that these findings appear to be in contradiction to previous studies (e.g. Ruiter et al. mention that “most authors of case reports report multiple pregnancies as a risk indicator for VP; however, our data fail to support this finding”). As such, we have included some additional wording to the relevant sentence, to specify that the focus on singleton pregnancies is a simplifying assumption based on findings indicating no significant association between multiple pregnancies and VP incidence.

Question/Comment: The UK NSC estimate of the incidence of VP of 0.03% of pregnancies is likely an underestimation. The authors have addressed this. However, the UKOSS may have its own inherent flaws.

Response: Many thanks for this observation. We fully agree with the stated issue of obtaining robust relevant incidence estimates for this rare condition to use as input in the model. However, we hope that this is sufficiently accounted for by the consideration of relevant scenario analyses, as well as the already included discussion of the general parameter uncertainty (pointing out the prevalence of VP in particular) and the UKOSS study specifically (in the context of diagnostic criteria and the possible discrepancy between ultrasound-detected and clinically presenting VP).

Question/Comment: An important finding is that clearly using IVF alone as the screening for vasa previa will detect few cases, and lead to the highest number of deaths from VP.

Response: Many thanks, we agree with this comment and the assessment of the results for the IVF-based strategy in particular.

 

Journal Requirements

Question/Comment: Please ensure that your manuscript meets PLOS ONE's style requirements, including those for file naming.

Response: We can confirm that, to our understanding, the manuscript adheres to the relevant style requirements.

Question/Comment: Please expand the acronym “UK NSC” (as indicated in your financial disclosure) so that it states the name of your funders in full. Please state what role the funders took in the study. If the funders had no role, please state: "The funders had no role in study design, data collection and analysis, decision to publish, or preparation of the manuscript." If this statement is not correct you must amend it as needed. This information should be included in your cover letter; we will change the online submission form on your behalf.

Response: Please find the correspondingly updated funding statement below.

“This study was funded by the UK National Screening Committee (UK NSC); the author group included employees (JM, CV, AM) and members (CH) of the Committee, or its subgroups (BT, ORA), who contributed to the study design, data collection and analysis, decision to publish, and preparation of the manuscript. The views and opinions expressed by the authors in this publication are not necessarily those of the UK NSC.” 

Question/Comment: Please confirm that the statements provided in the Competing Interests sections do not alter your adherence to all PLOS ONE policies on sharing data and materials, by including the following statement: "This does not alter our adherence to PLOS ONE policies on sharing data and materials.” If there are restrictions on sharing of data and/or materials, please state these. Please note that we cannot proceed with consideration of your article until this information has been declared. 

Response: Please find the correspondingly updated Competing Interests statement below.

“All authors have completed the ICJME uniform disclosure form at www.icmje.org/coi_disclosure.pdf (available on request from the corresponding author) and declare: no support from any organisation for the submitted work other than that described above; JM, CV, and AM are employees of the UK NSC secretariat which funded the submitted work; CH is a member of the UK NSC; BT and ORA are members of the Fetal, Maternal and Child Health Group (FMCH) of the UK NSC; GA is a Council member of the Royal College of Obstetricians and Gynaecologists and a Steering Committee member of the UK Obstetric Surveillance System; BRF, JK and THC are, or were formerly, employed by Costello Medical which was commissioned for the model development and supportive work by the UK NSC; no other relationships or activities that could appear to have influenced the submitted work.

The above statement does not alter our adherence to PLOS ONE policies on sharing data and materials. ” 

Question/Comment: In your Data Availability statement, you have not specified where the minimal data set underlying the results described in your manuscript can be found. PLOS defines a study's minimal data set as the underlying data used to reach the conclusions drawn in the manuscript and any additional data required to replicate the reported study findings in their entirety. All PLOS journals require that the minimal data set be made fully available. For more information about our data policy, please see http://journals.plos.org/plosone/s/data-availability.

Response: We can confirm that the Microsoft Excel-based model (suitable for replicating all results presented in the submitted manuscript) has now been made available and uploaded as supporting information (S4 File: Model spreadsheet).

Question/Comment: Please review your reference list to ensure that it is complete and correct. If you have cited papers that have been retracted, please include the rationale for doing so in the manuscript text, or remove these references and replace them with relevant current references. Any changes to the reference list should be mentioned in the rebuttal letter that accompanies your revised manuscript. If you need to cite a retracted article, indicate the article’s retracted status in the References list and also include a citation and full reference for the retraction notice.

Response: We can confirm that the list of references is still complete and correct; an additional reference (Bobrowska et al. 2022) has been included in response to one of the peer review comments and other reference information (e.g. hyperlinks) have been updated where required.

---

## [Decision Letter · Decision Letter 1]

20 Oct 2022

PONE-D-21-26780R1The impact of ultrasound-based antenatal screening strategies to detect vasa praevia in the United Kingdom: an exploratory study using decision analytic modelling methodsPLOS ONE

Dear Dr. Ruban-Fell,

Thank you for submitting your manuscript to PLOS ONE. After careful consideration, we feel that it has merit but does not fully meet PLOS ONE’s publication criteria as it currently stands. Therefore, we invite you to submit a revised version of the manuscript that addresses the points raised during the review process.

ACADEMIC EDITOR: The initial submission was only reviewed by a single reviewer, myself. Consequently, when the revised version was submitted, it was decided to send it out for review by 2 new reviewers. The reviews are favorable overall, and we anticipate that subject to satisfactorily addressing the concerns of reviewer 2, the manuscript should be acceptable for publication.I do have a few concerns. 1. You have referred to the Australia and New Zealand guidelines. These guidelines do recommend identification of placental cord insertion where feasible. Kindly address this in your revision. Please see the attached file and also the link below.https://onlinelibrary.wiley.com/doi/pdf/10.1002/sono.12222 2. Reference 18 is an oral abstract reference. To my knowledge, this data was never published. Please only include published or accepted manuscripts as references.==============================

Kind regards,

Yinka Oyelese

Guest Editor

PLOS ONE

Journal Requirements:

Reviewers' comments:

Reviewer's Responses to Questions

**Comments to the Author**

Reviewer #1. My comments from the previous review have been adequately and satisfactorily addressed.

Reviewer #2: All comments have been addressed

Reviewer #3: (No Response)

2. Is the manuscript technically sound, and do the data support the conclusions?

Reviewer #2: Partly

Reviewer #3: Partly

3. Has the statistical analysis been performed appropriately and rigorously? 

Reviewer #2: Yes

Reviewer #3: Yes

4. Have the authors made all data underlying the findings in their manuscript fully available?

Reviewer #2: Yes

Reviewer #3: Yes

5. Is the manuscript presented in an intelligible fashion and written in standard English?

Reviewer #2: Yes

Reviewer #3: Yes

6. Review Comments to the Author

Reviewer #2: The authors tackled very important topic of targeted screening and carried out an exploratory study using decision analytic modelling methods examining various targeted screening methods for vasa previa in the United Kingdom. They find that examining VCI based screening and low-lying placentas resulted in the highest detection rates and lowest rates of perinatal death.

Although I am not sure regarding the strength of the methodology (although it sounds) this well written manuscript. The conclusions are supported by the methods. However, I agree with other reviewer comment that any model is only as good as the assumptions put into it.. Furthermore, the authors do not examine the role of universal screening and another a flaw of this study is the question: “What constitutes “screening” for vasa previa….

Neve the less the authors address the remarks and flaws raised. This is interesting, well written and important study theretofore I recommend accept it.

Reviewer #3: The authors of this UK based study explore targeted screening based on IVF pregnancy, low lying placenta and velamentous cord insertion, compared to no screening. They conclude, using a decision analytical tree method, that screening for VP based on those pregnancies affected by low lying placenta is potentially the most efficient method, avoiding false positive diagnoses and having least impact on the current care pathway.

In Lines 93-94 they state that “The overall purpose of this work was to make a practical contribution to the evolving discussion about the antenatal detection of VP in the UK; this was achieved by presenting here an analysis of four possible detection pathways for VP”. This the authors have done using generally reasonable assumptions. However, I am struck that the screening/diagnostic phase of ascertainment of VP in this study is assumed to be by TAS at 32 weeks, when cord insertion and the presence of low lying placenta can be determined with a high degree of certainty at the mid trimester scan (as discussed in reference 7 that the authors cite). Hence what is missing from this analysis is a policy of referral for TAS/TVS at 32 weeks where there is LLP or VCI at the routine mid trimester scan. This would dramatically reduce the number of additional 32 week scans required while very modestly increasing the length of a mid trimester scan and only in those units that don’t already look for cord insertion, as all do already for LLP.

Lines 99-100: The VP screening model was programmed in Microsoft Excel and used a decision-analytic tree structure to explore the effects of four potential detection pathways in a hypothetical one-year UK pregnancy cohort.

-What was the assumed size of this cohort? I cannot see this defined anywhere, except in table 3 862, 785 pregnancies are referred to.

Lines 124-126: In the VCI-based pathway, additional testing for VCI and BL/S placenta specifically aimed at establishing the risk of VP would be performed during the 18+0 to 20+6 week scan, with positive detection prompting a recall at 32 weeks to perform further TAS to confirm the presence of VP (representing a population screening strategy based on a risk factor which is sought for the sole purpose of and preventing adverse outcomes from VP, and which is not currently reported in UK practice).

-it is not clear what is the screening process: is it a TA scan at 18-20 weeks, followed by a further TAS at 32 weeks, and then a TVS as the diagnostic test? Sorry if I have missed this but if it is submerged in the text somewhere it needs to be clearer, possibly by means of a flow diagram. If the authors are suggesting (as I state above) that LLP and VCI screening can be carried out at 18-20+6 weeks, this would hugely reduce the estimate of additional 32 week TAS scans referred to in Table 3 in the VCXI pathway.

Lines 132-133: In all pathways, pregnancies that underwent TAS at 32 weeks were also followed-up by TVS for VP, if VP could not be excluded. Incidental detection of VP across all pregnancies was also accounted for in all four pathways.

-TAS cannot exclude VP, TVS can. The normal pathway is TAS for screening and TVS for diagnosis, I am curious to know why the authors suggest that this should be inverted? In other words, once VP is suspected then a TVS should be performed, not another TAS, which would in any event require another TVS.

Line 149: ….workshops involving six UK clinical experts (GA, BT, NT, AM, EDJ, HG).

-How were the clinical experts chosen?

Lines 325-330. The base case results of this exploratory study showed that the modelled VCI-based and LLP-based pathways led to the detection of a greater proportion of VP pregnancies and a higher number of referrals to TVS than the no screening or IVF-based pathways. These higher VP detection rates also led to a correspondingly lower proportion of VP pregnancies resulting in perinatal death. The VCI-based pathway resulted in the highest VP detection rate (78.9%) and lowest proportion of perinatal death in VP pregnancies (14.2%); however, it also resulted in the detection of almost all VCI pregnancies, compared to minimal detection of VCI in the other pathways, and required a substantially higher number of additional TAS scans which, currently, are rarely recommended in practice.

-is TVS the mode of diagnosis, in which case why are additional TAS referenced?

Lines 356-360: This case-fatality rate is informed by the literature on clinically presenting VP and is also aligned with the conclusions made from a UK single-centre study by Zhang et al. where the authors estimated that around half of the 21 ultrasound-detected cases of VP (in a cohort of 26,830 pregnancies) would have resulted in stillbirth if they had not been diagnosed prenatally.[2, 20] However, these estimates of VP-related mortality contrast with a 2017 national clinical surveillance study conducted in the UK where, in a cohort of approximately 750,000 pregnancies, six deaths to VP were reported.[18]

-It is true that the estimates of VP and mortality associated with it vary. The authors reference a UKOSS study (ref 18) which is an abstract and contains incomplete information. It is unusual that the UKOSS study on VP appears never to have been published, and this limits the robustness of the estimates given. Nevertheless UKOSS is a surveillance system and very likely under-estimates the true incidence of VP. The authors should comment on this point.

Table 3: The premise that an additional TAS (presumably at 32 weeks) would be required in 862,785 women to screen for VCI need some explanation. Many UK units routinely report cord insertion-this takes around 30 seconds as part of the 18-20 week (or indeed 11-14 week) scan.

---

## [Author Response · Author response to Decision Letter 1]

1 Dec 2022

Editor

Question/Comment: You have referred to the Australia and New Zealand guidelines. These guidelines do recommend identification of placental cord insertion where feasible. Kindly address this in your revision.

Response: As suggested, we have now included additional wording in the Introduction to make explicit reference to the RANZCOG guideline, as an example of available guidance in the context of VCI as possible candidate for population screening. 

Further to this, we would like to highlight again the already included reference to the RANZCOG guideline as part of the Discussion, when mentioning the existence of guidance that does indeed recommend VCI-based screening.

Question/Comment: Reference 18 is an oral abstract reference. To my knowledge, this data was never published. Please only include published or accepted manuscripts as references.

Response: We agree with the Editor that the full results of the UKOSS study of vasa praevia do not appear to have been published to date; as such, we would suggest the following amendments to the manuscript:

- Instead of referring to the abstract when discussing the increasing interest in this area as part of the Introduction, we have now included a reference to the UKOSS 2016 annual report which features a section on vasa praevia

- In acknowledgment of the fact that the previously included UKOSS scenario analysis was based on unpublished data obtained through personal communication with the principal investigator (GA), we have now excluded this scenario from the manuscript

- We would suggest to still include the original reference to the abstract when using this to highlight the difference in reported VP mortality rates in the Discussion, as we consider this to be an important point in the discussion of screening and the UKOSS-related statement to be sufficiently supported by the data from the abstract (as published in a supplementary issue of the BJOG; https://obgyn.onlinelibrary.wiley.com/doi/10.1111/1471-0528.14585) – however, in order to acknowledge the nature of the reference, we have also included additional wording to highlight that only preliminary results from the UKOSS study are currently available

Reviewer 1

Question/Comment: The authors tackled very important topic of targeted screening and carried out an exploratory study using decision analytic modelling methods examining various targeted screening methods for vasa previa in the United Kingdom. They find that examining VCI based screening and low-lying placentas resulted in the highest detection rates and lowest rates of perinatal death.

Although I am not sure regarding the strength of the methodology (although it sounds) this well written manuscript. The conclusions are supported by the methods. However, I agree with other reviewer comment that any model is only as good as the assumptions put into it.. Furthermore, the authors do not examine the role of universal screening and another a flaw of this study is the question: “What constitutes “screening” for vasa previa….

Neve the less the authors address the remarks and flaws raised. This is interesting, well written and important study theretofore I recommend accept it.

Response: Thank you very much for your positive feedback and recommendation.

Reviewer 2

Question/Comment: The authors of this UK based study explore targeted screening based on IVF pregnancy, low lying placenta and velamentous cord insertion, compared to no screening. They conclude, using a decision analytical tree method, that screening for VP based on those pregnancies affected by low lying placenta is potentially the most efficient method, avoiding false positive diagnoses and having least impact on the current care pathway.

In Lines 93-94 they state that “The overall purpose of this work was to make a practical contribution to the evolving discussion about the antenatal detection of VP in the UK; this was achieved by presenting here an analysis of four possible detection pathways for VP”. This the authors have done using generally reasonable assumptions. However, I am struck that the screening/diagnostic phase of ascertainment of VP in this study is assumed to be by TAS at 32 weeks, when cord insertion and the presence of low lying placenta can be determined with a high degree of certainty at the mid trimester scan (as discussed in reference 7 that the authors cite). Hence what is missing from this analysis is a policy of referral for TAS/TVS at 32 weeks where there is LLP or VCI at the routine mid trimester scan. This would dramatically reduce the number of additional 32 week scans required while very modestly increasing the length of a mid trimester scan and only in those units that don’t already look for cord insertion, as all do already for LLP.

Response: We agree with the Reviewer’s thoughts on the likely merit of including VCI or LLP as part of a possible screening strategy, and we believe this to indeed be covered by the modelled strategies.

Namely, the LLP-based screening strategy involves the routinely performed diagnosis of LLP at the mid-trimester scan as basis for the subsequent referral for TAS/TVS for VP at 32 weeks. Similarly, the VCI-based pathway includes the additional testing for VCI at the mid-trimester scan to indicate the referral for TAS/TVS for VP at 32 weeks in cases where VCI is detected at 18+0 to 20+6 weeks.

In this context, we would also like to highlight Figure 1 in the submission which provides a flowchart detailing the different steps of the modelled screening pathways, in line with the description above. 

Question/Comment: Lines 99-100: The VP screening model was programmed in Microsoft Excel and used a decision-analytic tree structure to explore the effects of four potential detection pathways in a hypothetical one-year UK pregnancy cohort.

-What was the assumed size of this cohort? I cannot see this defined anywhere, except in table 3 862, 785 pregnancies are referred to.

Response: The overall cohort size (i.e. 862,785 pregnancies), which had so far been included in the supplementary Table S1, has now also been added to the list of key inputs in Table 2.

Question/Comment: Lines 124-126: In the VCI-based pathway, additional testing for VCI and BL/S placenta specifically aimed at establishing the risk of VP would be performed during the 18+0 to 20+6 week scan, with positive detection prompting a recall at 32 weeks to perform further TAS to confirm the presence of VP (representing a population screening strategy based on a risk factor which is sought for the sole purpose of and preventing adverse outcomes from VP, and which is not currently reported in UK practice).

-it is not clear what is the screening process: is it a TA scan at 18-20 weeks, followed by a further TAS at 32 weeks, and then a TVS as the diagnostic test? Sorry if I have missed this but if it is submerged in the text somewhere it needs to be clearer, possibly by means of a flow diagram. If the authors are suggesting (as I state above) that LLP and VCI screening can be carried out at 18-20+6 weeks, this would hugely reduce the estimate of additional 32 week TAS scans referred to in Table 3 in the VCI pathway.

Response: We would like to highlight again the flowchart that has been provided as part of Figure 1 in the submission. In line with this, we can confirm that the VCI-based pathway does indeed include proposed testing for VCI (in addition to the routine testing for LLP) as part of the mid-trimester scan at 18+0 to 20+6 weeks; as described above, detected cases of VCI would then be referred to TAS for VP at 32 weeks (with a final confirmatory TVS for TAS-detected VP at this stage).

We can further confirm that the stated number of referrals to 32-week TAS (for VP) in the VCI-based pathway are modelled as a direct result of the literature-informed incidence of VCI as well as the expected TAS test accuracy for VCI at 18+0 to 20+6 weeks.

Question/Comment: Lines 132-133: In all pathways, pregnancies that underwent TAS at 32 weeks were also followed-up by TVS for VP, if VP could not be excluded. Incidental detection of VP across all pregnancies was also accounted for in all four pathways.

-TAS cannot exclude VP, TVS can. The normal pathway is TAS for screening and TVS for diagnosis, I am curious to know why the authors suggest that this should be inverted? In other words, once VP is suspected then a TVS should be performed, not another TAS, which would in any event require another TVS.

Response: We agree with the Reviewer regarding the appropriate sequence of screening steps, which we have also further clarified for the modelled strategies in response to the queries above, and apologise for any confusion stemming from the use of “cannot be excluded” for describing the referral to subsequent screening steps.

In response, we have amended the wording where relevant (e.g. as part of the flowchart in Figure 1) to instead say that confirmatory/diagnostic TVS was performed where VP “was suspected” during TAS.

Question/Comment: Line 149: ….workshops involving six UK clinical experts (GA, BT, NT, AM, EDJ, HG).

-How were the clinical experts chosen?

Response: We have now added an additional sentence to the Methods section detailing how workshop participants were selected (based on their previous and/or current interaction with the UK NSC).

Question/Comment: Lines 325-330. The base case results of this exploratory study showed that the modelled VCI-based and LLP-based pathways led to the detection of a greater proportion of VP pregnancies and a higher number of referrals to TVS than the no screening or IVF-based pathways. These higher VP detection rates also led to a correspondingly lower proportion of VP pregnancies resulting in perinatal death. The VCI-based pathway resulted in the highest VP detection rate (78.9%) and lowest proportion of perinatal death in VP pregnancies (14.2%); however, it also resulted in the detection of almost all VCI pregnancies, compared to minimal detection of VCI in the other pathways, and required a substantially higher number of additional TAS scans which, currently, are rarely recommended in practice.

-is TVS the mode of diagnosis, in which case why are additional TAS referenced?

Response: As outlined in the clarification of the modelled screening strategies further above, TVS for VP does indeed represent the final, diagnostic step of each pathway. Additional scans in the form of TAS have also been considered as intermediate steps of the different pathways, in order to explore potential resource implications of the different strategies, for example where additional scans would be required over and above what is expected for current clinical practice (please also refer to our response to the corresponding query further below).

Question/Comment: Lines 356-360: This case-fatality rate is informed by the literature on clinically presenting VP and is also aligned with the conclusions made from a UK single-centre study by Zhang et al. where the authors estimated that around half of the 21 ultrasound-detected cases of VP (in a cohort of 26,830 pregnancies) would have resulted in stillbirth if they had not been diagnosed prenatally.[2, 20] 

However, these estimates of VP-related mortality contrast with a 2017 national clinical surveillance study conducted in the UK where, in a cohort of approximately 750,000 pregnancies, six deaths to VP were reported.[18]

-It is true that the estimates of VP and mortality associated with it vary. The authors reference a UKOSS study (ref 18) which is an abstract and contains incomplete information. It is unusual that the UKOSS study on VP appears never to have been published, and this limits the robustness of the estimates given. Nevertheless UKOSS is a surveillance system and very likely under-estimates the true incidence of VP. The authors should comment on this point.

Response: The manuscript offers variation in diagnostic criteria as a potential explanation for the differences between VP-related mortality estimates, and we consider this particularly plausible when comparing estimates derived from screening studies (such as Zhang et al.) and those from observation of clinically presenting VP (such as the UKOSS study). 

Additional wording has now been added to acknowledge the possibility of under-reporting in the UKOSS study. However we consider it important to note, in the discussion with the Reviewer, that the UKOSS system is well known to UK obstetricians and that, when presenting clinically, VP is a condition which is difficult to miss; as such, we do not think that under-reporting should be overemphasised.

Further to this, and also in response to the Editor’s request above, we have included mention of the information from UKOSS being based on currently available, preliminary results, in order to acknowledge that the source of the information is indeed a conference abstract.

Question/Comment: Table 3: The premise that an additional TAS (presumably at 32 weeks) would be required in 862,785 women to screen for VCI need some explanation. Many UK units routinely report cord insertion-this takes around 30 seconds as part of the 18-20 week (or indeed 11-14 week) scan.

Response: We can confirm that the model currently assumes that additional TAS for VCI would be required for all pregnancies, in addition to the routine fetal anomaly scan at 18+0 to 20+6 weeks, as, to our understanding, the detection of VCI is not formally mandated by the scan base menu detailed in the NHS fetal anomaly screening programme handbook, or in NICE or RCOG guidance. 

However, we acknowledge that this is likely a conservative assumption based on the fact that some UK units may indeed routinely report cord insertion, and have added further wording to this effect in the Discussion.

---

## [Editor Report · Decision Letter 2]

4 Dec 2022

The impact of ultrasound-based antenatal screening strategies to detect vasa praevia in the United Kingdom: an exploratory study using decision analytic modelling methods

PONE-D-21-26780R2

Dear Dr. Ruban-Fell,

We’re pleased to inform you that your manuscript has been judged scientifically suitable for publication and will be formally accepted for publication once it meets all outstanding technical requirements.

Kind regards,

Yinka Oyelese

Guest Editor

PLOS ONE

---

## [Editor Report · Acceptance letter]

8 Dec 2022

PONE-D-21-26780R2 

The impact of ultrasound-based antenatal screening strategies to detect vasa praevia in the United Kingdom: an exploratory study using decision analytic modelling methods 

Dear Dr. Ruban-Fell:

I'm pleased to inform you that your manuscript has been deemed suitable for publication in PLOS ONE. Congratulations! Your manuscript is now with our production department. 

Kind regards, 

on behalf of

Dr. Yinka Oyelese 

Guest Editor

PLOS ONE